# HCC-Related lncRNAs: Roles and Mechanisms

**DOI:** 10.3390/ijms25010597

**Published:** 2024-01-02

**Authors:** Mimansha Shah, Devanand Sarkar

**Affiliations:** 1Department of Human and Molecular Genetics, Virginia Commonwealth University, Richmond, VA 23298, USA; shahmj3@vcu.edu; 2Department of Human and Molecular Genetics, Massey Comprehensive Cancer Center, and VCU Institute of Molecular Medicine (VIMM), Virginia Commonwealth University, Richmond, VA 23298, USA

**Keywords:** hepatocellular carcinoma, long noncoding RNA, prognosis, treatment

## Abstract

Hepatocellular carcinoma (HCC) presents a significant global health threat, particularly in regions endemic to hepatitis B and C viruses, and because of the ongoing pandemic of obesity causing metabolic-dysfunction-related fatty liver disease (MAFLD), a precursor to HCC. The molecular intricacies of HCC, genetic and epigenetic alterations, and dysregulated signaling pathways facilitate personalized treatment strategies based on molecular profiling. Epigenetic regulation, encompassing DNA methyltion, histone modifications, and noncoding RNAs, functions as a critical layer influencing HCC development. Long noncoding RNAs (lncRNAs) are spotlighted for their diverse roles in gene regulation and their potential as diagnostic and therapeutic tools in cancer. In this review, we explore the pivotal role of lncRNAs in HCC, including MAFLD and viral hepatitis, the most prevalent risk factors for hepatocarcinogenesis. The dysregulation of lncRNAs is implicated in HCC progression by modulating chromatin regulation and transcription, sponging miRNAs, and influencing structural functions. The ongoing studies on lncRNAs contribute to a deeper comprehension of HCC pathogenesis and offer promising routes for precision medicine, highlighting the utility of lncRNAs as early biomarkers, prognostic indicators, and therapeutic targets.

## 1. Introduction

Hepatocellular carcinoma (HCC) is the most common primary liver cancer arising from hepatocytes. It is the sixth most common cancer and the third most common cause of cancer-related deaths globally according to Global Cancer Incidence, Mortality and Prevalence (GLOBOCAN) 2020 [1]. It develops predominantly in the context of chronic liver disease and is characterized by its aggressive nature and limited treatment options. HCC is a significant global health concern with considerable regional variation. The highest incidence rates are observed in regions where HCC-causing chronic hepatitis B virus (HBV) and hepatitis C virus (HCV) infections are endemic, such as Eastern and South-Eastern Asia (Mongolia, Thailand, Cambodia and Vietnam), Northern and Western Africa (e.g., Egypt and Niger), and sub-Saharan Africa [1]. Other significant risk factors include alcoholic liver disease; non-alcoholic fatty liver disease (NAFLD), which has been renamed metabolic dysfunction-associated fatty liver disease (MAFLD); and aflatoxin exposure [1,2]. Recent studies emphasize the increasing prevalence of HCC in Western countries due to the rising rates of MAFLD [3]. Because of the diverse nature of its etiology, understanding the molecular mechanisms underlying HCC has been a major focus of research.

Genetic and epigenetic alterations, including mutations in genes such as *TP53*, *CTNNB1*, and *TERT*, are common in HCC [4]. The dysregulation of key signaling pathways, such as the Wnt/β-catenin, MAPK, and PI3K-AKT pathways, plays a critical role in HCC development and progression [5]. Recent research has elucidated the intricate crosstalk among these pathways and their potential as therapeutic targets [5]. The roles of noncoding RNAs (ncRNAs), epigenetic modifications, and microenvironmental factors in shaping the HCC landscape are being uncovered, offering novel targets for intervention [6,7]. The early diagnosis of HCC is crucial for improving patient outcomes. Surveillance strategies include the use of imaging modalities such as ultrasound, computed tomography (CT), and magnetic resonance imaging (MRI), coupled with alpha-fetoprotein (AFP) measurements. Advances in imaging techniques have improved early detection and staging accuracy [8]. Liquid biopsy approaches, such as circulating tumor DNA (ctDNA) analysis, are also being explored for their potential to detect HCC-related mutations and biomarkers [9].

HCC management depends on the stage at diagnosis and the patient’s overall health [10]. Surgical resection, liver transplantation, and locoregional therapies, like radiofrequency ablation (RFA) and transarterial chemoembolization (TACE), remain crucial for early-stage HCC [11]. Immune checkpoint inhibitors, such as anti-PD-1 antibodies nivolumab and pembrolizumab, have shown promise in advanced HCC treatment [12]. Currently, the most effective FDA-approved treatment for advanced non-resectable HCC is a combination of anti-PD-L1 (atezolizumab) and anti-VEGF (bevacizumab) antibodies, providing an overall response rate of 27% [13,14,15]. Additionally, targeted therapies, including tyrosine kinase inhibitors (TKIs), like sorafenib and lenvatinib, continue to be integral in systemic HCC treatment [16]. Prevention strategies are vital in HCC management. Vaccination against HBV, treatment of HCV infection, and lifestyle modifications to reduce MAFLD risk are essential preventive measures [10,17]. Future research efforts focus on developing innovative therapies, identifying early biomarkers, and refining surveillance strategies to further improve HCC outcomes.

## 2. Epigenetic Regulation

Epigenetic regulation represents a critical layer of gene expression control that extends beyond the genetic code. It encompasses a range of molecular processes that modify the chromatin landscape and thereby govern gene activity [18]. Epigenetic mechanisms play a fundamental role in cellular differentiation, development, and adaptation to environmental stimuli. Epigenetic regulation in HCC involves a complex interplay of molecular mechanisms that control gene expression and function without altering the underlying DNA sequence. These epigenetic modifications can significantly impact the development and progression of HCC.

### 2.1. DNA Methylation: Beyond the Methyl Mark

DNA methylation, characterized by the addition of methyl groups to cytosine residues, is a hallmark of epigenetic regulation. Recent research has unveiled the intricate dynamics of DNA methylation patterns in response to developmental cues and environmental factors [19]. Methylation patterns exert direct control over gene expression by influencing transcriptional accessibility and ultimately shaping cellular phenotypes [20]. The hypermethylation of promoter regions in tumor suppressor genes, such as *CDKN2A* (p16) and *APC*, can lead to their inactivation [21]. Conversely, global hypomethylation in HCC can contribute to genomic instability and tumorigenesis [22].

### 2.2. Histone Modifications: The Chromatin Code

Histone modifications, including acetylation, methylation and phosphorylation, serve as the chromatin code that regulates gene accessibility and transcription. Advances in chromatin biology have illuminated the intricate interplay among various histone marks, revealing their roles in orchestrating gene expression programs [23]. Recent studies have emphasized the significance of histone modifications in cellular plasticity and disease processes [24]. Aberrant histone modifications can lead to changes in chromatin structure and gene accessibility, impacting gene expression patterns and resulting in HCC [25].

### 2.3. Noncoding RNAs: Epigenetic Mediators

In higher eukaryotes, the genome is pervasively transcribed to generate a large number of noncoding RNAs (ncRNAs), particularly long noncoding RNAs (lncRNAs) and microRNAs (miRNAs), which have emerged as crucial epigenetic mediators. They participate in gene silencing, chromatin remodeling, and post-transcriptional regulation [26]. Recent investigations have unraveled the functional diversity of ncRNAs, shedding light on their roles in cellular homeostasis and pathology [27]. The dysregulation of ncRNAs, including miRNAs, such as miR-21 and miR-122, and lncRNAs, is a hallmark of HCC regulating the development and progression of the disease [28]. The field of lncRNA is growing at a very fast speed, unraveling novel mechanisms of action. In this review, we aim to provide a comprehensive and up-to-date description of the role of lncRNAs in hepatocarcinogenesis, highlighting new definitions of lncRNAs, novel mechanisms of action, potential use as diagnostic and prognostic marker, and targets for therapeutic intervention.

## 3. Introduction to lncRNAs: Mechanisms and Modes of Action

LncRNAs have taken the center stage of molecular biology research, with a surge in investigations uncovering their intricate roles in diverse cellular processes. LncRNAs are typically defined as transcripts exceeding 200 nucleotides (nt) without significant protein-coding potential. However, a recent consensus statement recommends the definition of lncRNAs as ncRNAs greater than 500 nt in length and mainly generated by RNA polymerase II (Pol II), to distinguish them from transcripts generated by RNA polymerase III (Pol III) and some Pol II transcripts, such as snRNAs and intron-derived snoRNAs, which are ~50–500 nt in size [29]. Additionally, lncRNAs coding for biologically active micropeptides are increasingly being identified [30]. Thus, the definition of lncRNAs is undergoing continuous evolution.

There are multiple classification systems for lncRNAs encompassing various categories, such as length of transcript, genomic location and context, sequence and structure conservation, effects on DNA sequences, functional mechanisms and targeting mechanisms, and association with protein coding genes or subcellular structures [31,32]. In relation to the sites of transcription relative to annotated protein-coding genes, lncRNAs are classified as sense, antisense, bidirectional, intronic, and intergenic (Figure 1) [33]. Contemporary studies have illuminated the vast functional repertoire of lncRNAs. These molecules play pivotal roles in gene regulation at both transcriptional and post-transcriptional levels. They participate in chromatin remodeling, epigenetic modifications, and subcellular compartmentalization, exerting influence over fundamental cellular processes and pathways [34]. Understanding the underlying mechanisms and modes of action by which lncRNAs exert their influence is fundamental to unraveling their significance in cellular biology.

### 3.1. Gene Expression Regulation

One of the primary mechanisms through which lncRNAs operate is the regulation of gene expression. They can modulate transcriptional processes by interacting with chromatin, acting as scaffolds for transcriptional complexes, or influencing the recruitment of transcription factors (Figure 2). lncRNAs also regulate gene expression at post-transcriptional and translational levels.

#### 3.1.1. Regulation of Chromatin

RNAs have inherent chromatin regulatory function because the negatively charged RNAs can neutralize positively charged histone tails, resulting in the opening up of chromatin structure in a rapid manner [35]. LncRNAs are similarly involved in the complex regulation of chromatin architecture either by directly binding to DNA or by interacting with DNA-binding and RNA-binding proteins in either a *cis* or a *trans* manner.

##### LncRNA–Protein Interaction Mediating Chromatin Regulation

Homeodomain transcription factors Hox maintain positional identity, and lncRNA HOTTIP is transcribed from the 5′ end of HOXA locus in an antisense direction [36]. HOTTIP binds to the WD repeat domain 5 (WDR5) protein, a core component of the MLL/SET1 histone methyltransferase complex, and guides this complex to specific genomic loci, including the HOXA gene cluster. This results in the trimethylation of histone H3 at lysine 4 (H3K4me3) and the activation of HOXA genes, providing an example of the *cis* regulation of gene expression by a lncRNA [36].

Conversely, lncRNAs can interact with chromatin modifiers, sequestering them from promoters of specific genes. Histone deactylase SIRT6 inhibits many pluripotency-associated genes, and lncPRESS1 interacts with and sequesters SIRT6, thereby maintaining the acetylation of histone H3 at lysine 56 and lysine 9 (H3K56ac and H3K9ac) in the promoters of these genes and facilitating the pluripotency of embryonic stem cells (ESCs) [37]. p53 induces the expression of developmental genes and the differentiation of ESCs by inhibiting lncPRESS1 expression, which frees up SIRT6, allowing it to deacetylate and inhibit the expression of pluripotency genes [37].

Polycomb Repressive Complex 2 (PRC2) is an evolutionarily conserved gene-silencing machinery inducing the trimethylation of histone H3 at lysine 27 (H3K27me3), resulting in epigenetic silencing and the transcriptional suppression of target genes. PRC2 complex contains multiple proteins, including EZH2 and SUZ12, and many lncRNAs interact with these proteins recruiting PRC2 to specific genomic sites and causing methylation-induced transcription repression. One such example is HOTAIR, which can recruit PRC2 to numerous sites, and the overexpression of HOTAIR in an epithelial cell shifts the PRC2 occupancy pattern to the embryonic fibroblast pattern, thereby facilitating invasion and metastasis by these cells [38]. The lncRNA ANRIL binds to CBX7 in PRC2 and mediates the CBX7-mediated suppression of cyclin-dependent kinase inhibitors *CDKN2A* and *CDKN2B*, thereby preventing cellular senescence [39].

##### LncRNA-DNA Interaction Mediating Chromatin Regulation

During transcription, lncRNAs can form RNA–DNA hybrids, such as R-loops, which can be recognized by chromatin modifiers or transcription factors to either activate or inhibit transcription. lncRNA TARID, generated from the antisense strand of the *TCF21* gene, forms an R-loop at the promoter of the transcription factor TCF21 [40]. GADD45A binds to this R-loop and recruits methylcytosine dioxygenase TET1, thereby facilitating the local demethylation of *TCF21* promoter and inducing its expression [40]. *TCF21* is a tumor-suppressor gene regulating the cell cycle, and the TARID-regulated mechanism allows *TCF21* expression in normal cells while both TARID and *TCF21* are hypermethylated, and hence downregulated, in cancer cells [41].

#### 3.1.2. Regulation of Transcription

LncRNAs regulate transcription in multiple ways, including interaction with a variety of transcription factors (TFs) either inhibiting or activating their functions [42]. lncRNA GAS5, expressed in growth-arrested cells, binds to the glucocorticoid receptor (GR), inhibiting the GR-dependent expression of apoptosis-inhibiting genes [43]. Brain-specific lncRNA RMST interacts with transcription factor SOX2, facilitating its DNA-binding and promoting neuronal differentiation [44]. CHASERR is an evolutionarily conserved lncRNA that is transcribed from near the transcription start site of the ATP-dependent chromatin-remodeling enzyme CHD2 [45]. CHASERR expression interferes with transcription factor accessibility to the CHD2 promoter, and CHASERR loss leads to increased CHD2 expression [45]. Homozygous Chaserr knockout mice are embryonically lethal, and heterozygotes show significant growth retardation [45].

### 3.2. Post-Transcriptional Regulation

One major mechanism by which lncRNAs exert post-transcriptional regulation is by interacting with protein complexes that regulate mRNA splicing and protein turnover. Serine/arginine (SR) splicing factors regulate tissue- and cell-type-specific alternative splicing (AS). MALAT1 is localized in nuclear paraspeckles, where it interacts with SR splicing factors, affects their distribution, and regulates pre-mRNA alternative splicing [46]. The depletion of MALAT1 causes nuclear fragmentation and aberrant mitosis, potentially because of the inclusion of an alternative exon in SAT1 pre-mRNA [46]. Some lncRNAs harbor specific motifs through which they interact with splicing factors; e.g., lncRNA PNCTR has YUCUYY and YYUCUY motifs, through which it interacts with PTBP1, thereby inhibiting the splicing of pre-mRNAs that bear the same motifs [47].

Certain lncRNAs act as competitive endogenous RNAs (ceRNAs) or sponges, sequestering microRNAs (miRNA) and preventing their interaction with target mRNAs. These lncRNAs are abundant and harbor miRNA-complementary sites. ceRNA function has been attributed to numerous lncRNAs, and this regulatory network has been shown to play a role in cancer progression [48]. lncRNA PNUTS have seven binding sites for miR-205, which targets ZEB1 and ZEB2. PNUTS overexpression sponges miR-205, and subsequent increases in ZEB1 and ZEB2 contribute to epithelial–mesenchymal transition (EMT) and invasion by breast cancer cells [49].

### 3.3. Liquid–Liquid Phase Separation (LLPS)

LLPS inside a cell involves the polymerization of macromolecules, creating unique liquid/fluid phases without any delimiting membrane, similar to an oil droplet in water, which can form or disassemble in seconds, thus quickly and accurately responding to signals and stimuli [50]. Subcellular compartments, such as nucleoli and stress granules, are generated by LLPS, and interaction between RNA-binding proteins (RBPs) containing intrinsically disordered regions (IDRs) and RNAs drives the phase-separation reaction, leading to the formation of ribonucleoprotein (RNP) granules [51].

LncRNAs can function as molecular scaffolds to orchestrate phase separation and facilitate the assembly of membrane-free organelles. lncRNA levels are low and often cancer-specific, with their expression being induced by extracellular cues during cancer development and progression [52]. LncRNAs participate in LLPS via their repetitive sequences, which contribute to molecular crowding and allowing specific cell states and cancer phenotypes [52]. Nuclear paraspeckle, which is rich in RNA and protein, is an example of a membrane-less organelle generated by LLPS, and lncRNA NEAT1 (Nuclear paraspeckle assembly transcript 1) plays an essential role in nuclear paraspeckle formation [53]. NEAT 1 has two transcripts; the longer transcript NEAT1_2 interacts with RBP NONO (non-POU domain-containing octamer binding protein), SFPQ (splicing factor proline and glutamine-rich), and FUS (FUS RNA-binding protein), facilitating paraspeckle formation [54]. Paraspeckles, and hence NEAT1, play a role in the nuclear retention of mRNAs, function as molecular sponges for RBPs, and regulate key physiological and pathological processes, such as corpus luteum formation, and cancer [55,56].

Additional examples of lncRNAs functioning through LLPS are provided in Table 1.

### 3.4. lncRNAs Coding Micropeptides

Although lncRNAs are by definition ncRNAs, recent studies have shown that some lncRNAs indeed code for functional micropeptides. One of the earliest papers describing this phenomenon was published in 2020, describing the expression of a 60-amino-acid peptide ASRPS from LINC00908, which is downregulated in triple-negative breast cancer (TNBC) [68]. ASRPS binds to STAT3, inhibits STAT3 phosphorylation, and prevents angiogenesis [68]. LINC00665 codes a 52-amino-acid peptide CIP2A-BP, the translation of which is inhibited by TGF-β in TNBC cells [69]. CIP2A-BP inhibits the oncogene CIP2A, thereby inhibiting multiple oncogenic signaling pathways and CIP2A-BP-abrogated lung metastases in the MMTV-PyMT mouse breast cancer model [69]. lncAKR1C2 is an exosomal lncRNA secreted from gastric cancer cells, and it encodes a micropeptide pep-AKR1C2, which is produced in lymphatic endothelial cells and promotes lymph node metastasis by gastric cancer cells by modulating YAP phosphorylation and increased fatty acid β-oxidation [70]. lncRNA AC115619 was shown to code for the micropeptide AC115619-aa, which was downregulated in human HCC and inhibited HCC progression [71]. AC115619-aa interacted with WTAP and inhibited the assembly of N6-methyladenosine (m6A) methyltransferase complex, thereby inhibiting global m6A levels in human HCC cells and affecting the expression of tumor-associated proteins SOCS2 and ATG14 [71]. The number of micropeptides encoded by lncRNAs is growing, especially in the context of cancer. However, for most of these micropeptides, there is a single article describing their detection and function, and in some cases, they are detected in non-human systems [72]. As such, further in-depth validation studies are required to confirm the generalizability of these observations.

## 4. Significance of Studying lncRNAs in HCC

Studying lncRNAs in HCC holds paramount significance in advancing our understanding of this challenging cancer and improving clinical outcomes. Recent research has underscored the multifaceted roles of lncRNAs in HCC, revealing their potential as critical diagnostic, prognostic, and therapeutic tools (Figure 3).

First and foremost, the identification of lncRNAs as early diagnostic markers for HCC has garnered significant attention. Studies have pinpointed specific lncRNAs, such as HULC (Highly Upregulated in Liver Cancer) and HEIH (hepatocellular carcinoma UpRegulated EZH2-Associated Long Noncoding RNA), with elevated expression in HCC patients. These lncRNAs serve as promising candidates for early biomarkers, allowing for timely intervention and improved patient outcomes [73,74]

LncRNAs have also demonstrated substantial prognostic value in HCC. For instance, HOTAIR, known for its association with poor overall survival and disease-free survival in HCC patients, offers crucial insights for tailoring treatment strategies and predicting patient outcomes [75]. Other lncRNAs, such as H19, have also been linked to HCC progression and poor prognosis, emphasizing the prognostic potential of these molecules [76].

Furthermore, the intricate roles of lncRNAs in HCC pathogenesis have come to light. lncRNAs, including MALAT1, have been implicated in promoting aggressive phenotypes and tumor progression. Additionally, lncRNAs play a pivotal role in epigenetic regulation by modulating DNA methylation and histone modifications, thereby influencing gene expression patterns in HCC cells [77]. Understanding these lncRNA-driven epigenetic mechanisms offers insights into HCC pathogenesis and potential therapeutic interventions.

Importantly, lncRNAs are emerging as potential therapeutic targets in HCC. Recent research has explored innovative strategies to target lncRNAs, including the use of antisense oligonucleotides and small molecules. These approaches hold promise for disrupting critical pathways implicated in HCC development and progression [78]. Additionally, the exploration of lncRNA-directed therapies offers a new dimension to the treatment of this challenging cancer.

## 5. LncRNAs in NASH and Viral Hepatitis, Risk Factors Contributing to HCC

LncRNAs have emerged as critical players in various liver diseases, including non-alcoholic steatohepatitis (NASH) and viral hepatitis, which are precursors to HCC. A number of lncRNAs have been identified from mouse models of NASH. However, for some of these lncRNAs, the human homologs either do not exist or have not yet been identified. A discussion of the mouse lncRNAs, which lack human relevance, has been omitted from this review.

### 5.1. NASH

NASH is characterized by hepatic inflammation and fibrosis, often associated with obesity and metabolic syndrome. Recent research has unveiled the regulatory roles of several lncRNAs in NASH pathogenesis. For instance, the lncRNA H19 has garnered attention for its contribution to hepatic fibrosis in NASH. H19 is thought to act as a ceRNA, sponging miRNAs and thereby modulating the expression of genes involved in fibrosis [79]. Through this mechanism, H19 promotes the activation of hepatic stellate cells and the deposition of extracellular matrix components, exacerbating fibrosis progression. MALAT1 was induced in hepatocytes by palmitate treatment as well as in ob/ob mice [80]. It was documented that MALAT1 interacted with SREBP1c mRNA to increase its stability, and in vivo MALAT1 siRNA injection prevented hepatic lipid accumulation and insulin resistance in ob/ob mice [80].

Additionally, lncRNAs like MEG3 have been implicated in NASH-associated oxidative stress and inflammation. MEG3 acts as a molecular sponge for miR-34a, alleviating its inhibitory effect on SIRT1, a key regulator of cellular responses to stress. By enhancing SIRT1 expression, MEG3 attenuates oxidative stress and inflammation in the liver, potentially mitigating NASH development [81].

### 5.2. Viral Hepatitis

Viral hepatitis, particularly hepatitis B (HBV) and hepatitis C (HCV), is a major risk factor for the development of chronic liver diseases, including cirrhosis and HCC. Emerging evidence suggests that lncRNAs play crucial roles in the host–virus interactions and the progression of viral hepatitis.

The lncRNA HULC has been linked to HBV replication and hepatocarcinogenesis. HULC modulates viral replication by interacting with host factors and promoting the expression of critical viral genes. Furthermore, it contributes to HCC by enhancing the stability of SIRT1 mRNA and inhibiting its degradation by miR-372, leading to increased SIRT1 expression and subsequent tumor growth [81].

In the case of HCV, the lncRNA MALAT1 has been implicated in regulating HCV replication. MALAT1 interacted with HCV-encoded proteins, facilitating viral genome replication [82]. This interaction underscores the intricate interplay between lncRNAs and viral replication processes.

miR-675 is derived from the first exon of H19, and it was shown that H19 and miR-675 were upregulated in HBV-induced chronic hepatitis with the subsequent downregulation of miR-675 target PPARα with associated activation of Akt/mTOR signaling and perturbation of energy metabolism [83].

## 6. LncRNAs Regulating HCC

A PubMed search performed on 12 December 2023 with the key words lncRNA and HCC revealed 2327 papers, of which 224 were review papers, clearly demonstrating a growing interest in studying lncRNA in hepatocarcinogenesis. However, many of these studies are in vitro using only one cell line; some studies are focused on intrahepatic cholangiocarcinoma, not HCC; and some studies unraveled mouse lncRNAs without identifying the corresponding human homologue. The abundance of lncRNA is an important consideration for determining its mechanism of action and validating whether a particular lncRNA can really modulate a phenotype. Usually, the lncRNA level is low, which might be adequate to modulate the function of a single target-gene locus, especially for *cis*-acting lncRNAs. However, to sequester a protein or a protein complex or to function as a ceRNA for miRNAs, high levels of lncRNAs are required. It was shown that the lncRNA PNCTR can sequester only ~7.12–27.44% of its target protein PTBP1 in HeLa cells because of the relative abundance of each [47]. However, this level of inhibition was sufficient for PNCTR to regulate cell survival via its regulation of PTBP1 [47]. Given this constraint, stoichiometry analysis, as well as in-depth molecular analysis by multiple techniques, are necessary to determine whether a lncRNA–protein or a lncRNA–miRNA interaction identified by pull-down assays can actually have an effect on biological function and alter a phenotype. The primary mechanism of action for many of the lncRNAs in HCC was shown to be sponging miRNAs, and with the caveat described above, the role of these lncRNAs in HCC requires stringent validation. Here, we highlight those lncRNAs whose function has been interrogated in multiple studies with reproducible results (Table 2), or in the case of a single study, the experiments were performed rigorously using multiple cell lines and in vivo models.

### 6.1. LncRNAs Functioning as Oncogenes

#### 6.1.1. HOTAIR

HOTAIR is a 2158-nucleotide-long lncRNA that was identified from a custom tilling array of the HOXC locus (12q13.13) [38]. The upregulated expression of HOTAIR in HCC patients correlated with significantly lower cumulative recurrence-free survival [104,105]. As described in Section 3, HOTAIR modulates PRC2, resulting in epigenetic silencing and the transcriptional suppression of target genes [96,106]. Interaction between HOTAIR and EZH2, a component of PRC2, lead to a decrease in miR-218 levels and an increase in its target BMI1, which functions as an oncogene [105]. In HepG2 and Bel7404 cells, HOTAIR knockdown activated p14^ARF^ and p16^Ink4a^ signaling, causing cell cycle arrest and inhibiting the growth in xenograft models [105]. HOTAIR’s competitive binding to RNA helicase DDX5 displaced RNA-binding E3 ligase MEX3B, which stabilized SUZ12, a core subunit of PRC2, and promoted PRC2-mediated gene silencing, especially the PRC2 target genes *EpCAM* and pleuripotency genes [107]. Liver tumors from HBV X protein (HBx) and c-Myc transgenic mice and chronically HBV-infected patients showed a negative correlation between DDX5 levels, pleuripotency gene expression, and HCC differentiation [107]. According to these findings, a potential role of HOTAIR in the negative regulation of HBV/HCC is suggested, which is contrary to the oncogenic function of HOTAIR and needs further verification.

HOTAIR was shown to regulate exosome production by HepG2 cells by modulating expression and localization of exosomal proteins, such as RAB35, SNAP23, and VAMP3, and through its ability to interact with RAB35 protein [108]. Exosomes are known to promote metastasis. However, it was not studied whether HOTAIR-mediated increase in exosome production accentuates metastasis.

In HCC, HOTAIR disrupted the expression and function of SETD2, influencing histone modifications and DNA repair, which promoted the growth of cancer stem cells (CSCs) [109]. HOTAIR expression is regulated by the transcription factor FOXC1, and it interacts with miR-1, miR-145, miR-122, and RNA-binding motif protein 38 (RBM38) in HepG2, Bel-7402 and Huh7 cells, all contributing to HCC [110,111,112,113].

#### 6.1.2. MALAT1

MALAT1 is a highly conserved nuclear-localized lncRNA transcribed from human chromosome 11q13 as a ~7.5 kb transcript. It is overexpressed in primary tumors and metastases and is prone to copy number changes in several cancer types, including HCC [114]. In HCC patients, it is associated with metastasis and poor prognosis [77].

Nuclear paraspeckles are important structures that retain specific mRNAs in the nucleus [115]. MALAT1 is localized in nuclear paraspeckles, where it interacts with serine/arginine (SR) splicing factors, affects their distribution, and regulates pre-mRNA alternative splicing [46]. In HCC cells, MALAT1 upregulated the splicing factor SRSF1, which caused alternative splicing of RPS6KB1 with subsequent activation of mTORC1, a process that induced the transformation of liver progenitor cells [116]. MALAT1 was shown to activate the Wnt/β-catenin pathway, although the underlying molecular mechanism was not elucidated [116]. MALAT1 was shown to promote glycolysis and suppress gluconeogenesis by increasing the translation of the transcription factor TCF7L2 [117]. However, the mechanism by which MALAT1 increased TCF7L2 translation was not clear because a direct interaction between MALAT1 and TCF7L2 was not examined.

MALAT1 can act as a ceRNA, sequestering miRNAs and preventing them from targeting their downstream mRNA targets. In HCC cells, such as Bel-7402, Hep3B, HepG2, HuH-7, MHCC97, and SMMC-7721, MALAT1 sponges miR-195 causing the activation of EGFR, PI3K/AKT, and JAK/STAT signaling; miR143-3p upregulating ZEB1; miR-146-5p upregulating TNF receptor-associated factor 6 (TRAF6) resulting in activation of AKT; miR-22 increasing snail family transcription factor SNAIL; and miR-30a-5p upregulating vimentin, all of which contribute to augmentation of cell proliferation, invasion, and EMT [118,119,120,121,122].

A recent study unraveled a role of MALAT1 in mitochondria of HCC cells, such as HepG2 and HL7702, where it interacted with multiple loci in mitochondrial DNA (mtDNA), such as D-loop, *COX2*, *ND3*, and *CYTB* genes [123]. Knocking down MALAT1 perturbed mitochondrial transcription, resulting in the inhibition of mitochondrial functions, such as oxidative phosphorylation and ATP production, a decrease in mitophagy, and an increase in apoptosis [123]. It was shown that the translocation of MALAT1 from the nucleus to the mitochondria is mediated by RNA-shuttle protein HuR and mitochondrial membrane protein MTCH2 [123].

#### 6.1.3. HULC

HULC, a 482 bp transcript encoded by a gene in chromosome 6p24.3, is a well-studied lncRNA that has garnered considerable attention in the context of HCC. HULC was first discovered as a highly upregulated transcript in HCC, and its aberrant expression has been associated with HCC development and progression [89]. One of its primary functions is to act as a molecular sponge for miRNAs, such as miR-2001-3p, miR-186, and miR-107, resulting in an increase in the oncogenes ZEB1 (contributing to EMT), HMGA2, and E2F1, respectively [124,125]. Transcription factor E2F1 regulates sphingosine kinase 1 (SPHK1), which is known to promote angiogenesis, and a contribution of HULC/E2F1/SPHK1 axis in augmenting tumor angiogenesis was shown [125]. Protective autophagy contributes to tumorigenesis, and SIRT1 has been identified to positively regulate this process. In HepG2 and Hep3B cells, HULC downregulates miR-6825-5p, miR-6845-5p, and miR-6886-3p, resulting in the upregulation of their target ubiquitin-specific peptidase 22 (USP22) and the inhibition of ubiquitin-mediated degradation of SIRT1 [81]. This mechanism contributed to HULC- and SIRT1-mediated protective autophagy [81]. HULC decreased miR-15a in Hep3B cells, which resulted in the inhibition of PTEN via P62, resulting in the activation of the oncogenic PI3K-AKT-mTOR pathway [126].

HULC is involved in regulating HBV-induced HCC. The HBX protein activated the HULC promoter via CREB, and the induced HULC inhibited the promoter of the tumor suppressor eukaryotic translation elongation factor 1 epsilon 1 (*EEF1E1*/P18), thus reducing its expression and facilitating HBX-augmented proliferation both in vitro and in vivo [127]. However, the mechanism by which HULC regulates EEF1E1 expression was not explored. Increased lipid production fuels tumor growth by providing a source of energy. In HepG2 and HuH-7 cells, HULC inhibited the miR-9 promoter by inducing CpG island methylation, leading to an increase in miR-9 target peroxisome-proliferator-activated receptor alpha (PPARA) and subsequently PPARA target acyl-CoA synthetase subunit *ACSL1*, which resulted in increased cholesterol production and stimulation of cell proliferation [128]. Conversely, exogenous cholesterol augmented the HULC promoter activity of retinoid x receptor (RXRA), thus establishing a positive feedback loop. HULC increased the expression of the circadian rhythm regulating gene *CLOCK* by interacting with 5′-UTR of the *CLOCK* gene, which was suggested to contribute to the in vivo growth of HepG2 cells [129]. Thus, HULC augments a variety of oncogenic pathways engaging diverse mechanisms to promote HCC.

#### 6.1.4. PVT1

Chromosome 8q amplification is a frequent event in many cancers, including HCC. PVT1 (Plasmacytoma Variant Translocation 1) is located on chromosome 8q24 adjacent to *MYC* (encoding c-MYC), and *MYC* and *PVT1* genes are co-amplified in multiple cancer patients, including a subset of HCC patients [130]. PVT1 is frequently upregulated in HCC tissues and cell lines, and its overexpression has been associated with aggressive clinicopathological features and poor prognosis in HCC patients [131,132]. It was demonstrated that PVT1 promotes cell proliferation and cancer stem-cell-like properties by binding to and stabilizing the RNA-binding protein NOP2 [132]. PVT1 functions as ceRNA for multiple miRNAs, thereby increasing their targets, such as miR-365 and ATG3, to promote autophagy, and miR-150 and hypoxia-inducible protein 2 (HIG2) to modulate iron metabolism and cell proliferation [133,134]. PVT1 has been shown to inhibit interferon-α-induced apoptosis in SMMC-7721 HCC cells by interacting with the signal transducer and activator of transcription 1 (STAT1) [135].

#### 6.1.5. HOTTIP

HOTTIP (HOXA Transcript at the Distal Tip) is a 7.9 kb lncRNA located in chromosome 7p15 that is frequently upregulated in HCC tissues and cell lines, and its overexpression is correlated with aggressive clinicopathological features, including advanced tumor stages and poorer prognosis for HCC patients [36,95]. As described in Section 3, HOTTIP activates *HOXA* genes, which promote HCC cell proliferation, migration, and invasion [36,95]. HOTTIP itself is targeted by tumor suppressor miRNAs, miR-192, and miRNA-240, and this regulation was shown to interfere with glutaminolysis by glutaminase GLS1 in HepG2 and SMMC7721 cells [136]. Glutaminolysis is a hallmark of cancer cells, and it was suggested that HOTTIP might promote HCC by regulating glutamine metabolism. LncRNA PAARH functions as an oncogene in HCC and was shown to mediate its effect by sponging multiple miRNAs, such as miR-6760-5p, miR-6512-3p, miR-1298-5p, miR-6720-5p, miR-4516, and miR-6782-5p, resulting in the upregulation of HOTTIP [137]. Thus, there are a number of indirect mechanisms by which HOTTIP might be upregulated, facilitating HCC development and progression.

#### 6.1.6. H19

H19 is located in chromosome 11p15 and is associated with genomic imprinting. The genomic location of H19 is near the insulin-like growth factor 2 (*IGF2*) gene; while IGF2 is expressed only from the paternally inherited chromosome, H19 is expressed from the maternally inherited chromosome [138]. H19 is a 2.5-kilobase-long RNA molecule that undergoes splicing and polyadenylation, but it does not contain the information to produce a protein. H19 is highly conserved and is expressed in various cell lineages during mammalian development.

Whether H19 functions as an oncogene or tumor suppressor gene is a debatable issue in other cancers, but in HCC, the current literature suggests it as an oncogene [76]. H19 is frequently upregulated in HCC tissues and cell lines, and its elevated expression has been linked to poor clinical outcomes in HCC patients [139]. One of the key mechanisms through which H19 promotes HCC progression is by sponging miR-193b with resultant upregulation of MAPK1, promoting EMT and the transformation of stem cells. In HepG2 cells, H19 expression was induced by tumor-associated macrophages, suggesting that inflammation might regulate H19 expression [139]. H19 was upregulated in HCC tumors generated upon the deletion of Transforming growth factor-β receptor 2 (TGFBR2) via the activation of the transcription factor SOX2, and H19 knockdown inhibited TGFBR2-deletion-induced HCC [140]. However, the H19 targets that mediate this effect were not identified in this study. A potential role of H19 in promoting bone metastasis of HCC has been documented, in which H19 interacted with protein phosphatase 1 catalytic subunit alpha (PPP1CA), which dephosphorylated p38 MAPK with subsequent downregulation of osteoclastogenesis inhibitory factor osteoprotegerin (OPG) [141]. This resulted in the activation of osteoclastogenesis and hence osteolytic bone lesions, a feature of HCC bone metastasis. Additionally, H19 sponged miRNA-200b-3p, resulting in the enhanced expression of ZEB1, which caused increased cell migration and invasion [141].

#### 6.1.7. NEAT1

The nuclear paraspeckle assembly transcript 1 (NEAT1) gene is located in chromosome 11q13.1 and gives rise to two transcripts, a 3.7 kb NEAT1v1 and a 23 kb NEAT1v2, the latter of which is necessary for the formation of nuclear paraspeckles [53,142]. NEAT1, along with MALAT1, was also shown to bind to active chromatin sites, identified by the procedure Capture Hybridization Analysis of RNA Targets (CHART), suggesting a potential role in regulating transcription [143].

NEAT1 functions as an oncogene in many cancers, including HCC, in which it is overexpressed [144,145,146,147]. In Hep3B and HepG2 cells, the inhibition of NEAT1 attenuates proliferation, migration, and invasion, and NEAT1 functions by sponging multiple miRNAs and increasing the levels of the targets of these miRNAs, such as miR-485 and STAT3, miR-204 and ATG3 (which increases autophagy), and miR-139-5p and TGF-β1 [145,146,148].

A recent study, however, unravels an opposite role of NEAT1, mainly its long isoform NEAT1v2/NEAT1_2, which is regulating cancer cell metabolism [149]. Cancer cells use aerobic glycolysis for its survival, a phenomenon known as the “Warburg Effect”. It was shown that in HCC cells, mTOR, which promotes THE Warburg Effect, negatively regulates NEAT1 transcription, thus suppressing paraspeckle biogenesis and liberating the RNA-binding proteins NONO and SFPQ [149]. These RNA-binding proteins bind to U5 in spliceosomes and promote mRNA splicing and the expression of glycolytic enzymes, thus stimulating aerobic glycolysis and HCC growth. The mTOR inhibitor Rapamycin was shown to exert its anti-cancer effect by perturbing this mTOR/NEAT1-mediated mechanism [149]. The authors also showed that NEAT1v2/NEAT1_2 was downregulated in HCC patients, which correlated with poor overall survival [149]. This study underlies the importance of analyzing transcript variants of lncRNAs and checking their functions in the context of cancer.

#### 6.1.8. HEIH

HEIH (hepatocellular carcinoma upregulated EZH2-associated long noncoding RNA), located in chromosome 5q35, is a 1.7 kb transcript that is overexpressed in HCC patients and negatively correlates with cumulative survival [150]. HEIH interacts with EZH2, resulting in increased binding of EZH2 to and H3K27 trimethylation of p16 promoter, downregulating the expression of this tumor suppressor [150]. The overexpression and knockdown of HEIH stimulated and inhibited, respectively, xenograft growths of Hep3B, HepG2, and HuH-7 cells [150]. HEIH was detected in serum and exosome and was suggested as a potential biomarker of HCV-associated HCC [151]. HEIH was shown to contribute to sorafenib resistance by sponging miR-98-5p and activating the PI3K/Akt pathway [152].

#### 6.1.9. SNHG6

SNHG6 (small nucleolar RNA host gene 6) is another lncRNA that is located in chromosome 8q13.1, thus being amplified in many cancers, including HCC [153]. There are five transcripts of SNHG6 (SNHG6-003 to SNHG6-007), among which SNHG6-003 and SNHG6-006 were shown to be highly expressed in 52 HCC patients [153]. Mechanistically, it was shown that SNHG6-003 promoted HCC by sponging miR-26a/b and increasing the expression of its target transforming growth factor-β-activated kinase 1 (TAK1) [153]. In HCC cells, SNHG6 functions as ceRNA for multiple miRNAs inducing their targets, such as miR-101-3p and ZEB1, miR-139-5p and SERPINH1, let-7c-5p and MYC, and miR-6509-5p and HIF1A, all contributing to hepatocarcinogenesis [154,155,156,157].

A recent study linked SNHG6 in the progression of NAFLD to HCC by modulating cholesterol metabolism [158]. SNHG6 expression was induced by cholesterol treatment in HepG2 cells, and SNHG6 facilitated cholesterol-mediated interaction of ER-anchored FAF2 (Fas-associated factor family member 2) with mTORC1 at lysosomes promoting mTORC1-mediated increase in cellular cholesterol biosynthesis, thus establishing a positive feedback loop [158]. Snhg6 expression was induced in a high-fat/high-cholesterol (HF/HC) and diethylnitrosamine (DEN) model of mouse NAFLD/HCC; adeno-associated virus (AAV8)-mediated delivery of SNHG6 accelerated the process, and AAV8 delivering shRNA for SNHG6 reversed the process [158].

#### 6.1.10. Additional lncRNAs Functioning as Oncogenes

The oncogenic function of several lncRNAs in HCC has been described in a single albeit detailed study, which requires additional validation and/or characterization. Here, we highlight some of these lncRNAs affecting specific aspects of HCC development and progression.

##### LncRNAs Modulating Wnt/β-Catenin Signaling

LncRNA UFC1 was identified to be overexpressed in HCC tissues, and in vitro and in vivo studies unraveled a tumor-promoting role of UFC1 [159]. Mechanistically, UFC1 was shown to interact with and stabilize HuR mRNA with a resultant increase in β-catenin [159]. lncRNA for β-catenin methylation (lnc-β-Catm) was upregulated in liver cancer stem cells (CSCs) and induced the methylation of β-catenin by associating with β-catenin and methyltransferase EZH2 [160]. The methylation of β-catenin protected it from ubiquitination and increased its stability, leading to the activation of Wnt/β-catenin signaling that promoted the self-renewal of liver CSCs [160]. A novel mechanism of lncRNA action was proposed for DANCR (differentiation antagonizing non-protein coding RNA), in which it interacted with β-catenin protein and prevented β-catenin degradation by miR-214, miR-320a, and miR-199a, which resulted in increased stem cell line features in HCC cells [161].

##### LncRNAs Promoting HCC by Regulating Metabolism

LINC01234, overexpressed in HCC and correlated with poor prognosis, promoted HCC cell proliferation, migration, and drug resistance in vitro and in vivo [162]. It was shown that LINC01234 interacted with the promoter region of arginosuccinate synthase 1 (*ASS1*), inhibiting its transcription, especially by p53, thus modulating aspartate metabolism [162]. lncRNA RP11-386G11.10, transcriptionally regulated by ZBTB7A, functions as a ceRNA for miR-345-3p, increasing the expression of HNRNPU and its downstream lipogenic enzymes, with resultant accumulation of lipids in HCC cells and the promotion of metastasis and primary tumor growth [163]. A search for RNA-binding proteins (RBPs) regulating HCC identified that RBP CCT3 regulated lipid metabolism in HCC cells by regulating the lncRNA LINC00326, and interference with CCT3/LINC00326 interaction inhibited the growth of HuH-7 tumors implanted in zebrafish [164]. Taurine upregulated gene 1 (TUG1) interacted with PRC2 complex and epigenetically silenced a number of tumor suppressor genes, including p21. In HCC cells, TUG1-mediated downregulation of p21 resulted in upregulation of miR-455-3p with the resultant downregulation of its target AMPKβ2 (adenosine monophosphate-activated protein kinase subunit beta 2) [165]. This resulted in the activation of the mTOR pathway and the induction of hexokinase 2 (HK2), contributing to increased glycolysis, cell growth, and metastasis [165]. TUG1 was also shown to epigenetically silence the transcription factor KLF2, a tumor suppressor, in HCC cells [166]. Similar epigenetic silencing of KLF2 was also shown for lncRNA CDKN2B antisense RNA 1/ANRIL [167].

##### lncRNAs Regulating EMT and Metastasis

LncRNA-activated by TGF-β (lncRNA-ATB), overexpressed in HCC metastases and associated with poor prognosis, sponged miR-200 family, thereby upregulating ZEB1 and ZEB1 and inducing EMT [168]. Additionally, lncRNA-ATB bound to IL-11 mRNA, increasing its expression and activating STAT3, which facilitated the colonization of metastatic cells to distant organs [168]. lncRNA PRR34-AS1 sponged miRNA-296-5p to increase transcription factors E2F2 and SRY-box transcription factor 12 (SOX12) in HCC cells, activating Wnt/β-catenin pathway and promoting EMT [169]. It was also shown that HNRNPU augmented ZBTB7A expression, thus establishing a positive feedback loop [163]. lncRNA ZFAS1 (ZNFX1 antisense RNA1) levels correlated with intra- and extrahepatic HCC metastasis and poor prognosis, and it promoted HCC progression by squelching miRNA-150 with resultant upregulation of ZEB1 and matrix metalloproteinases MMP14 and MMP16 [170]. Extrahepatic metastatic tissues in HCC patients showed increased alternative splicing, which was associated with increased expression of DEAD-box RNA helicase 17 (DDX17) [171]. It was shown that DDX17 induced the retention of intron 3 of the lncRNA Paxillin antisense RNA 1 (PXN-AS1), creating a new transcript PXN-AS1-IR3 [171]. PXN-AS1-IR3 stimulated HCC metastasis by recruiting TEX10 (testis expressed 10) and p300 to MYC enhancer region and upregulating MYC levels. PXN-AS1 was increased in the serum of HCC patients with extrahepatic metastasis potentially serving as a blood biomarker [171]. In another study, splicing factor MBNL3 was shown to induce exon 4 inclusion of PXN-AS1, and this new transcript interacted with PXN mRNA, protecting it from miR-24-mediated degradation and stimulating HCC progression [172].

##### lncRNAs Regulating Vascular Invasion

LncRNA MVIH (lncRNA associated with microvascular invasion in HCC) was shown to be associated with increased microvascular invasion and metastasis and decreased recurrence-free survival in 215 HCC patients [173]. MVIH promoted angiogenesis and metastasis by increasing the secretion of phosphoglycerate kinase 1 (PGK1) [173]. lncRNA PAARH exerts its HCC-promoting effect in multiple ways, such as upregulation of HOTTIP by squelching multiple miRNAs and binding to HIF-1α and facilitating its recruitment to VEGF promoter, thereby increasing microvessel density and promoting angiogenesis and metastasis [137]. A major complication of HCC is portal vein tumor thrombus (PVTT), and it was shown that lncRNA ICR (ICAM-1-related) was expressed in Intercellular adhesion molecule 1 (ICAM-1)-positive cancer stem cells (CSCs) regulating CSC function and leading to PVTT development [174]. ICR was regulated by the stem cell transcription factor Nanog, and it positively regulated ICAM-1 levels by binding to ICAM-1 mRNA [174].

##### lncRNAs Regulating Oncogenic AKT Pathway

LncRNA cancer susceptibility 9 (CASC9) interacted with the RNA-binding protein heterogenous nuclear ribonucleoprotein L (HNRNPL), resulting in the activation of the AKT signaling pathway and contributing to increased tumorigenesis by HCC cells [175]. lncRNA RP11-295G20.2 interacted with the N-terminus of PTEN, facilitating its interaction with p62, translocation to lysosomes, and degradation, which led to the activation of the oncogenic AKT pathway [176].

##### Miscellaneous Functions of lncRNAs in HCC

The inhibition of the novel lncRNA NIHCOLE (noncoding RNA induced in hepatocellular carcinoma with an oncogenic role in ligation efficiency) induced apoptosis in multiple HCC cells with associated DNA damage because of a decrease in the nonhomologous end-joining (NHEJ) pathway of DNA double-strand breaks [177]. NIHCOLE was shown to associate with several NHEJ factors, including Ku70/Ku80 heterodimer, and promoted ligation efficiency [177]. CDIP transferase opposite strand, pseudogene (CDIPTOSP), also known as lnc-CTHCC, is a cancer testis gene that is highly expressed in the testis and HCC [178]. An lnc-CTHCC knockout mouse was protected from HCC development [178]. Lnc-CTHCC interacted with heterogenous ribonucleoprotein K (hnRNPK), recruiting it to the YAP1 promoter and activating its transcription [178]. Pluripotency and hepatocyte-associated RNA overexpressed in HCC (PHAROH) is a lncRNA that interacted with and sequestered the translation repressor TIAR, leading to increased translation of MYC, thereby promoting HCC [179].

### 6.2. lncRNAs Functioning as Tumor-Suppressor Genes

#### 6.2.1. MEG3

Maternally Expressed Gene 3 (MEG3) is located on chromosome 14q32. It is a ~1.6 kb transcript and is known for its tumor-suppressive functions [180]. MEG3 is downregulated in HCC tissues and cell lines, especially by the methylation of its promoter by DNA methyltransferases DNMT1 and DNMT2, and its reduced expression is associated with poor prognosis in HCC patients [103,181]. MEG3 upregulated p53 target genes by directly interacting with p53 DNA binding domain, and the overexpression of MEG3 in HepG2 and HuH-7 cells induced apoptosis [103,180]. MEG3 functions as ceRNA for many miRNAs, such as miRNA-664, but the functional implication of these findings needs further validation [103]. In HepG2 cells, MEG3 was shown to inhibit miR-10a-5p, which targets PTEN [182]. MEG3 overexpression induced PTEN and inhibited AKT signaling, and upregulated pro-apoptotic protein Bax and downregulated anti-apoptotic protein Bcl-2 [182]. An MS2 bacteriophage virus-like particles (VLPs) crosslinked with GE11 polypeptide was used to deliver MEG3 systemically, which inhibited xenografts of EGFR-positive HepG2 cells [183].

#### 6.2.2. GAS5

GAS5 (Growth Arrest-Specific 5), located in chromosome 1q25, is known for its role in regulating cell growth and apoptosis and is frequently dysregulated in cancers, including HCC, where its expression levels inversely correlate with patient survival [144,184]. The overexpression of GAS5 in HCC cells inhibited proliferation and invasion and inhibited vimentin via an unknown mechanism [184]. GAS5 levels are higher in sorafenib-resistant HCC cells and RNA-binding protein RBM38 was shown to bind to and stabilize GAS5 in these cells [185]. In HepG2 cells, GAS5 bound to glucose-regulated protein GRP78, and it induced apoptosis by activating the ER stress-signaling pathway [186]. In HepG2 and HuH-7 cells, GAS5 improved cisplatin sensitivity by sponging miR-222 [187].

#### 6.2.3. FENDRR

FOXF1-adjacent noncoding developmental regulatory RNA (FENDRR), located in chromosome 16q24, is downregulated in HCC and inhibits the growth of Hep3B and HepG2 cells in vivo [188]. It interacted with PRC2 and TrxG/MLL complexes, thus mediating the epigenetic silencing of gene expression [189]. FENDRR interacted with the promoter of Glypican-3 (GPC3), an HCC marker, and caused the methylation-induced silencing of GPC3 expression [188]. In MHCC97 cells, FENDRR sponged miR-423-5p, resulting in increased expression of its target growth arrest and DNA damage-inducible beta (GADD45B), leading to the suppression of in vivo tumorigenicity [190]. FENDRR has been suggested to regulate Tregs and immune escape, but these observations need further validation [190].

#### 6.2.4. Additional lncRNAs Functioning as Tumor Suppressors

Downregulated in liver cancer stem cells (DILC) is an ~2.4 kb lncRNA encoded by a gene located in chromosome 13q34 [191]. It is downregulated in liver CSCs, and its knockdown promotes in vivo tumorigenicity by liver CSCs [191]. DILC interacted with the IL-6 promoter, blocking NF-κB-mediated oncogenic IL-6/STAT3 signaling [191]. A recurrent deletion of lncRNA-PRAL (p53 regulation-associated lncRNA), located in chromosome 17p13.1, was identified in HCC patients [192]. The overexpression of lncRNA-PRAL induced apoptosis in HCCLM3 and SMMC-7721 HCC cells but not in Hep3B (p53-deficient) and HuH-7 (p53-mutant) cells, indicating that the tumor suppressor function of lncRNA-PRAL is mediated by p53 [192]. Three stem-loop motifs in the 5′ end of lncRNA-PRAL facilitated the interaction of HSP90 and p53, thereby precluding MDM2-mediated ubiquitination of p53 and increasing p53 stability [192]. The EZH2 transcript variant including exon 14 promotes metastasis. lncRNA LINC01348 was identified to be downregulated in HCC, and overexpression of LINC01348 inhibited in vivo metastasis by SK-Hep1 cells [193]. LINC01348 interacted with splicing factor 3b subunit 3 (SF3B3), interfering with exon 14’s inclusion of EZH2 and downregulating the expression of Snail [193]. lncRNA uc.134, downregulated in HCC, has been shown to interact with CUL4A, thus inhibiting ubiquitination of LATS1 and thereby silencing YAP [194]. lncRNA FTX, transcribed from X chromosome inactivation center, is downregulated in HCC patients and inhibits in vivo metastasis [195]. It sponges miR-374a, which facilitates the upregulation of genes that negatively regulate the Wnt/β-catenin pathway and binds to DNA replication factor MCM2, thereby inhibiting DNA replication [195].

## 7. lncRNAs Modulating Current HCC Treatment

For advanced HCC, the first line of treatment is a combination of immunotherapy and tyrosine kinase inhibitors (TKIs) such as sorafenib and lenvatinib. A number of in silico studies have identified potential lncRNA prognostic signatures determining responsiveness to TKIs and immunotherapy. Here, we highlight some of the lncRNAs that have been validated using in vivo studies to influence responsiveness and/or resistance to TKIs and modulate immunotherapy response.

Translation regulatory lncRNA 1 (TRERNA1) was upregulated by HBx in HCC cells that sponged miR-22-3p, resulting in the upregulation of NRAS and the activation of downstream RAF/MEK/ERK signaling, thereby contributing to sorafenib resistance [196]. A potential role of MALAT1 in sorafenib resistance, by sponging miR-140-5p and increasing its target Aurora-A kinase, has been described [197]. In HCC cells, the oncogene FOXM1 transcriptionally activated LINC-ROR (long intergenic non-protein coding RNA, regulator of reprogramming), which in turn increased FOXM1 levels by sponging miR-876-5p, thus establishing a positive feedback loop [198]. The upregulation of both FOXM1 and LINC-ROR was shown to confer resistance to sorafenib [198]. Sorafenib-resistant Hep3B and HuH-7 cells showed the enrichment of genes related to stemness and EMT, and single-cell RNA-sequencing in sorafenib-resistant HuH-7 cells identified the lncRNA ZFAS1 (ZNFX1 antisense RNA 1) as the highest upregulated transcript, which positively correlated with multiple stemness and EMT-associated genes in HCC patients [199]. Knocking down ZFAS1 by siRNA in sorafenib-resistant cells restored their sorafenib sensitivity [199]. However, the mechanism by which ZFAS1 regulates stemness genes was not explored in this study. lncRNA linc-VLDLR was shown to be secreted in the extracellular vehicles (EVs) by HepG2 and MzChA1 cells [200]. The treatment of human HCC cells with drugs, such as sorafenib, camptothecin, and doxorubicin, induced linc-VLDLR release in EVs, while incubation with the EVs protected cells from chemotherapy-induced cell death [200]. The knockdown of linc-VLDLR downregulated the drug transporter ABCG2 and ABCG2 overexpression rescued the effects of linc-VLDLR-mediated knockdown on increased sorafenib sensitivity [200]. The mechanism by which linc-VLDLR regulates ABCG2 remains to be explored. PHF8 is a histone lysine demethylase that functions as a transcriptional activator. lncRNA BBOX1-AS1 is upregulated in HCC patients, and it upregulated PHF8 by sponging miR-361-3p, which contributed to HCC progression, autophagy, and sorafenib resistance [201]. The mechanism by which PHF8 regulates these phenotypes remains to be determined. Sorafenib is known to induce ferroptosis in HCC cells. In sorafenib-resistant HCC cells, HIF1α induced the expression of lncRNA URB1-antisense RNA 1 (URB1-AS1), which contributed to ferroptosis via the phase separation of ferritin, and silencing URB1-AS1 restored sensitivity to sorafenib in an in vivo model [202]. A systematic bioinformatics analysis identified lncRNA LINC01132 as a potential oncogene for HCC [203]. It was shown that LINC01132 interacted with the transcription factor NRF1, leading to the induction of T cell activation antigen CD26/DPP4 [203]. Knocking down LINC01132 induced infiltration of CD8+ T cells and augmented the therapeutic efficacy of anti-PDL1 antibody in Hep1-6 cells, thereby suggesting a potential role of LINC01132 in regulating response to immunotherapy [203].

## 8. Conclusions and Future Perspectives

Through the exploration of various studies and research articles, it is evident that lncRNAs play crucial roles in the initiation, progression, and prognosis of HCC. These molecules have been found to be dysregulated in HCC, acting as both oncogenes and tumor suppressors, and their dysregulation can contribute to the altered expression of essential genes and signaling pathways involved in HCC development and pathogenesis. Furthermore, lncRNAs have exhibited potential diagnostic and prognostic value; e.g., circulating levels of lncRNA SCARNA10 (small cajal-body-specific RNA 10) have been shown to be higher in HBV- or HCV-positive HCC patients with positive correlation with tumor size, tumor stage, vascular invasion, and metastasis, highlighting their significance in clinical practice [204]. By analyzing the plasma or serum of cohorts of patients, several lncRNAs, including MALAT-1, HULC, and HOTAIR, have been implicated as biomarkers for HCC diagnosis but no comprehensive clinical trials are actively exploring these possibilities [205]. Therefore, further investigations are warranted to fully unravel the diverse mechanisms underlying lncRNAs’ involvement in HCC and to explore their therapeutic potential. Many HCC cell lines, such as BEL7402, SMMC7721, MHCC97L, BEL7404, QGY7701, QGY7703, QSG7701, and SKHEP1, used for lncRNA studies, have been found to be either contaminated or not of HCC origin [206]. Therefore, findings obtained from these cell lines must be further validated in additional cell lines. Overall, the body of knowledge garnered so far enhances our understanding of HCC biology and has promising implications for the development of innovative diagnostic methods and targeted therapies in the future. While 83.9% of human mRNAs are orthologous with mouse mRNAs, only 25% of human lncRNAs have mouse orthologs [207]. As such, studying the role of lncRNAs in HCC using mouse models is not always feasible. With ongoing advancements in high-throughput sequencing technologies and bioinformatics, the comprehensive understanding of lncRNAs in HCC is becoming more malleable. Thus, harnessing the potential of lncRNAs as both regulators and therapeutic targets in HCC holds immense promise in improving patient outcomes and revolutionizing the field of HCC and treatment.

Ongoing pre-clinical research explores the possibility of targeting lncRNAs using antisense oligonucleotides (ASOs), CRISPR-based strategies, and RNAi-based therapies. CRISPR and RNAi approaches have been tested mainly at the cell line level, where an oncogenic lncRNA was knocked out or knocked down by siRNA or shRNA, and then the tumorigenic potential of the cell line was checked in an in vivo model [88]. A recent in vivo genome-wide CRISPR activation screening using xenografts of MHCC-97H cells identified many HCC-promoting lncRNAs, and the clinical relevance of these lncRNAs to human HCC was validated [208]. This discovery approach can also be used therapeutically, where multiple HCC-promoting lncRNAs can be simultaneously knocked out with CRISPR/Cas9. Therapeutically, ASOs, which are short single-stranded DNAs, serve as a promising approach because of their ability to specifically hybridize with their target lncRNAs, thus forming DNA–RNA complexes, which can be recognized and degraded by RNase H. The intra-tumoral injection of ASO for MALAT1 significantly inhibited the growth of Hep3B and HuH-7 xenografts in nude mice [209]. HAND2-AS1 is a lncRNA shown to promote liver CSCs, and HAND-AS1 ASO could significantly inhibit the growth of HuH-7 and human patient-derived xenografts (PDX) in NOD-*Prkdc^scid^ Il2rg^tm1^*/Bcgen (B-NSG) mice [210]. On the other hand, PRAL is a tumor suppressor lncRNA, and overexpression of PRAL via an adenovirus inhibited xenograft growth of SMMC-7221 cells in nude mice [192]. These therapeutic approaches have been tested using subcutaneous xenografts in immunocompromised mice and intra-tumoral injection. Although these studies demonstrate a preliminary proof of principle, they do not determine whether the intravenous delivery of these approaches works in tumors in the liver and how an intact immune system responds to them. ASOs and similar oligonucleotide-based approaches have the potential to activate an interferon response, and viruses have the potential to develop neutralizing antibodies, thereby neutralizing their efficacy [211]. A potential strategy to overcome these drawbacks is to use targeted nanoparticles to deliver the payload. One example is the delivery of siRNA for the oncogene Astrocyte elevated gene-1 (AEG-1) via hepatocyte-targeted nanoparticles in an orthotopic model of HCC, as well as in a high fat diet-induced model of NASH showing marked therapeutic efficacy [212,213]. Similar strategies might be used for lncRNAs as well. As yet, no lncRNA-targeting therapeutics have entered clinical development, and more in-depth pre-clinical studies are required to bring the lncRNA-targeting approaches to the clinical arena. The field of lncRNA therapeutics is evolving rapidly, offering promising avenues for precision medicine [205]. In conclusion, the significance of lncRNAs as key regulators and potential targets in HCC is undeniably profound.

## Figures and Tables

**Figure 1 ijms-25-00597-f001:**
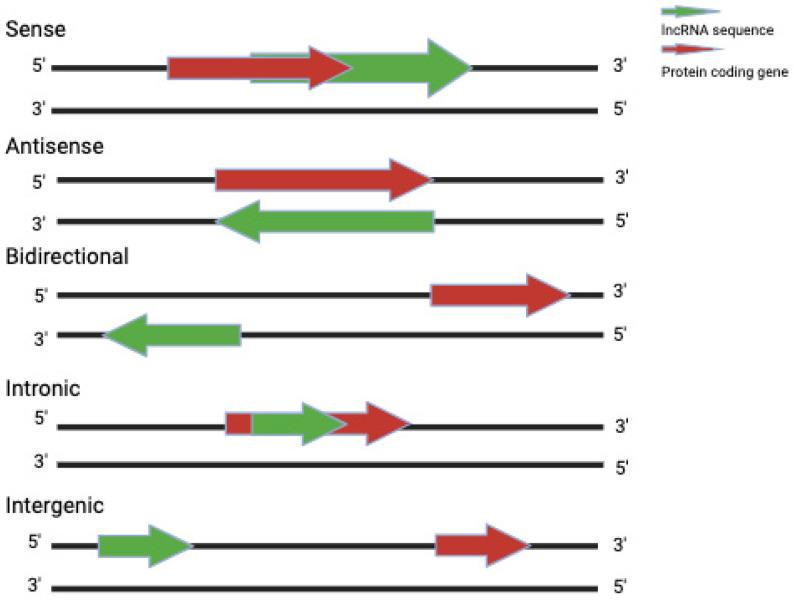
Different types of lncRNAs based on their genomic location. See text for details. Created in BioRender.com.

**Figure 2 ijms-25-00597-f002:**
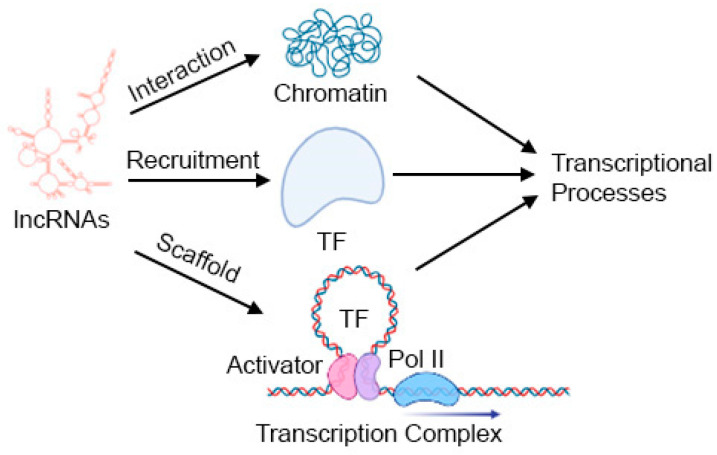
Mechanism of gene regulation by lncRNAs. TF: Transcription factor. See text for details. Created in BioRender.com.

**Figure 3 ijms-25-00597-f003:**
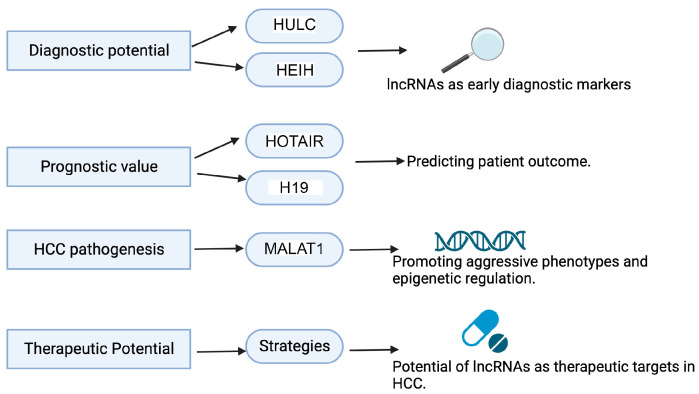
Significance of studying lncRNAs in HCC. See text for details. Created in BioRender.com.

**Table 1 ijms-25-00597-t001:** LncRNAs mediating their action by LLPS.

lncRNAs	Associated Protein	Mechanism	Phenotypes/Outcomes	References
DIGIT	BRD3	Forms phase-separated condensates of BRD3, facilitating the histone H3 acetylation of lysine 18 (H3K18ac) of enhancers of endoderm transcription factors	Endoderm differentiation	[57]
SLERT	DDX21	Formation of fibrillar center (FC) and dense fibrillar component (DFC) in the nucleolus	RNA polymerase I (Pol I) transcription and ribosomal RNA production	[58,59]
NORAD	PUM1 and PUM2	Sequestration of Pumilio proteins in PUM condensates termed NP bodies	Inhibition of Pumilio function, preventing aberrant mitosis and maintaining genomic stability	[60]
SNHG9	LATS1	LATS1 phase separation inhibits LATS1-mediated YAP phosphorylation	Activates YAP-driven gene transcription, thus promoting breast cancer	[61]
MELTF-AS1	YBX1	Phase separation of oncogenic RBP YBX1	Activation of ANXA8 transcription, leading to promotion of non-small-cell lung cancer (NSCLC)	[62]
LINP1	Ku70/Ku80	Multimerization of Ku to form filamentous Ku-containing aggregates	Facilitates Ku-mediated DNA repair by non-homologous end joining (NHEJ)	[63]
dilncRNA	DNA-damage-response (DDR) proteins, such as 53BP1	Molecular crowding of DDR proteins in LLPS condensates	Facilitates DNA double-strand break (DSB) repair	[64]
XiST	PTBP1, MATR3, TDP-43 and CELF1 RBPs	Forms a condensate in the inactive X (Xi)-compartment	X-chromosome inactivation	[65]
GIRGL	CAPRIN1	Sequesters CAPRIN1 and GLS1 mRNA in stress granules	Inhibition of GLS1 mRNA translation, facilitating the survival of cancer cells under glutamine-deprived conditions	[66]
NEAT1	TDP-43	Formation of nuclear bodies in response to stress	Mitigation of stress, the dysfunction of which might be a cause of ALS	[67]

**Table 2 ijms-25-00597-t002:** Role of dysregulated lncRNAs in HCC, which have been validated by multiple studies.

lncRNAs	Expression	Role in HCC	Outcome in HCC	References
HOTAIR	Upregulated	Oncogene	Promotes metastasis	[38,84,85]
MALAT1	Upregulated	Oncogene	Promotes proliferation, migration, invasion, and metastasis	[86,87,88]
HULC	Upregulated	Oncogene	Promotes proliferation and inhibits apoptosis	[85,89,90]
GAS5	Downregulated	Tumor Suppressor	Suppresses proliferation and invasion	[88,91]
PVT1	Upregulated	Oncogene	Promotes proliferation, migration, and invasion	[92,93,94]
HOTTIP	Upregulated	Oncogene	Promotes proliferation and angiogenesis	[95,96,97]
MVIH	Upregulated	Oncogene	Promotes proliferation, invasion, metastasis and angiogenesis	[98,99]
H19	Upregulated	Oncogene	Promotes proliferation, metastasis and angiogenesis	[99,100,101]
MEG3	Downregulated	Tumor suppressor	Inhibits proliferation, migration and invasion	[102,103]

## Data Availability

Not applicable.

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
