# Peer review of "HCC-Related lncRNAs: Roles and Mechanisms"

_ijms, 2024, doi:10.3390/ijms25010597_

Round 1
Reviewer 1 Report
Comments and Suggestions for Authors
The author presents a detailed investigation into the HCC related lncRNAs: roles and mechanisms. Upon thorough examination, it becomes evident that the review lacks a comprehensive focus on recent research developments and inadequately addresses its own research contributions. Moreover, the article fails to substantiate its relevance to the topic and primarily presents a summary of existing literature findings without delving into the requisite mechanistic intricacies.
My evaluation indicates that the article does not sufficiently satisfy the mechanistic depth required for comprehensive coverage in this field. Regrettably, based on these observations, I am unable to recommend the review article for publication in "International Journal of Molecular Sciences".
I encourage you to reconsider the review, focusing on incorporating recent advancements, emphasizing original research contributions, and delving deeper into the mechanistic underpinnings, thereby enhancing its suitability for publication.
Comments on the Quality of English Language
The review article exhibits a moderate level of English proficiency, but there are opportunities for enhancing clarity and precision within the language. Restructuring certain sentences could notably improve readability and coherence. I strongly advise conducting a comprehensive proofreading and revision process to specifically address these linguistic aspects.
Author Response
We thank the reviewers for their constructive criticisms and suggestions. We significantly modified the manuscript and re-written several sections to address the concerns of the reviewers. We believe the revised version of the manuscript is markedly improved. A point-by-point response is provided below:
Reviewer#1.
- The major concern of the reviewer is the lack of description of mechanism of action of lncRNAs. As the reviewer suggested, we expanded the mechanism of action markedly incorporating recent advancements, especially lncRNAs functioning in liquid-liquid phase separation and lncRNAs encoding micropeptides, in section 3 of the revised manuscript. We hope the reviewer recognizes our efforts and the revised content satisfies the reviewers’ concern.
- The reviewer asked for a comprehensive proofreading of the manuscript to improve the English language which we have meticulously performed.
Reviewer 2 Report
Comments and Suggestions for Authors
This review article by Shah and Sarkar described and summarized the roles and mechanisms of action of various long non-coding RNAs (lnRNAs) in contributing to hepatocellular carcinoma (HCC) pathogenesis. To my understanding, there are already several review articles published on the same topic. Please justify the difference and novelty of your review article.
I have also listed my major comments below.
1) The organization of this review article needs further improvement. Quite a few areas are redundant in terms of its content. For example, section 7 mechanism and mode of action of lncRNAs (7.1 -7.5) is very similar to section 4 introduction to lncRNAs. I suggest the authors combine these sections together. In another case, section 8.1 contains each individual lncRNA and their mode of action as subsections, which is fine. However, 8.1.10 “additional lncRNAs functioning as oncogene” is extremely long. Given the amount of description provided for these lncRNAs in this subsection, maybe the authors could consider separate these lncRNAs as previous subsections instead of combining them all together.
2) Some paragraphs do not contain descriptions that are detailed enough and as a result it reads like abrupt and random add up of sentences without much meaning or author’s intellectual input. Section 8.1.3 HULC, for example, has this problem. There are many more instances like this and I recommend the authors carefully go through the manuscript and paraphrase properly to make sure the manuscript reads smoothly.
3) Table 1 does not contain all lncRNAs that are discussed in the following section. Please complete the table.
4) The authors touched briefly on new therapeutic avenues targeting lncRNAs (line 164-166, 198-202). Could you expand and make this a new section to provide more details than just listing “antisense oligonucleotides, CRISPR-based strategies and RNA-based therapies”?
5) Figure 2, the positions of H19 and MALAT1 seem mixed up.
Minor.
1) Line 208, remove “the”
2) Line 370-371, “increase in the oncogene ZEB1” – is this an increase in the transcript or protein product?
3) Line 495, “than” change to “that”
Comments on the Quality of English Language
see minor comments above.
Author Response
We thank the reviewers for their constructive criticisms and suggestions. We significantly modified the manuscript and re-written several sections to address the concerns of the reviewers. We believe the revised version of the manuscript is markedly improved. A point-by-point response is provided below:
Reviewer#2.
- The major concern of the reviewer is organization of the manuscript and repetition of content. We meticulously addressed this issue. We incorporated the information in Section 2 of the original manuscript (Importance of understanding the molecular mechanisms underlying HCC) into Section 1 (Introduction) in the revised manuscript. We merged Section 4 (Introduction to long non-coding RNAs) and Section 7 (Mechanism and mode of actin of lncRNAs) of the original manuscript into Section 3 (Introduction to lncRNAs: mechanisms and modes of action) of the revised manuscript. We acknowledge that the section 8.1.10 (Additional lncRNAs functioning as oncogenes) in the original manuscript was too long. This section now is section 6.1.10 in the revised manuscript and we broke this section into subsections clustering the lncRNAs based on their mechanisms of action or based on the phenotypes they regulate. We hope that these modifications address the concern of the reviewers.
- Another concern is that some paragraphs do not contain detailed information without much meaning. We meticulously went over each section to make sure the information provided is coherent and meaningful. The inclusion of detailed mechanism of action in Section 3 of the revised manuscript will also help the clarification of mechanism of action in Section 6 better. We provided our own intellectual insight into the role of lncRNAs in HCC in the first paragraph of Section 6 (lncRNAs regulating HCC).
- Table 1 does not contain all the lncRNAs that are discussed. This table is now Table 2 in revised manuscript. In this table we highlight only those lncRNAs that have been validated and characterized by multiple in-depth studies.
- The reviewers asked for a separate section describing therapeutic avenues targeting lncRNAs. As yet no such avenue has moved into the clinical trial arena. As such we did not delve deep into this section. However, we added a new section, Section 7 (lncRNAs modulating current HCC treatment) in the revised manuscript, to describe how lncRNAs modulate tyrosine kinase inhibitors (TKIs) and immunotherapy, the first lines of treatment for HCC.
- The positions of H19 and MALAT1 were switched in Figure 3 of the revised manuscript. We apologize for the error.
Reviewer 3 Report
Comments and Suggestions for Authors
Authors are advised to revise the manuscript.

Comments on the Quality of English Language
Minor English corrections are needed.
Author Response
We thank the reviewers for their constructive criticisms and suggestions. We significantly modified the manuscript and re-written several sections to address the concerns of the reviewers. We believe the revised version of the manuscript is markedly improved. A point-by-point response is provided below:
Reviewer#3.
- The reviewer asked for more epidemiological information in various geographical regions based on GLOBOCAN 2020. The focus of the review paper is lncRNA not HCC epidemiology. The inclusion of information of each geographical region will make this section too long and distract the main focus of the paper. However, we provided the global information provided by GLOBOCAN 2020 in the Introduction (lines 29-31).
- We provided a list of abbreviations in Section 9 in the revised manuscript.
- The reviewer asked to describe the importance of the review paper at the end of the Introduction. We described the importance at the end of Section 2 in the revised manuscript to maintain the flow of the manuscript.
- We added future perspectives along with conclusions.
- We added an additional table, Table 1 in the revised version. The original Table 1 is now Table 2 in the revised version.
- We significantly modified the references to include more recent literature.
- We corrected the typos and capitalized LncRNA at the beginning of a sentence throughout the manuscript.
Round 2
Reviewer 1 Report
Comments and Suggestions for Authors
The author dedicated considerable effort to substantiate their findings and meticulously addressed all the reviewers' comments. Following the revision, this manuscript has significantly enhanced its quality. With no further modifications needed, the manuscript is now well-suited for publication in the International Journal of Molecular Sciences journal.
Author Response
We thank the reviewer for accepting our manuscript.
Reviewer 2 Report
Comments and Suggestions for Authors
In my previous comments, I suggested the authors to expand on the new therapeutic avenues targeting lncRNAs (which the authors mentioned in the first version of the manuscript line 164-166, 198-202). These included “antisense oligonucleotides, CRISPR-based strategies and RNA-based therapies” as the authors put in the text. They do not have to be those already in clinical trials. Any new therapeutics with potential to treat a disease, even at pre-clinical early discovery stage, is worth mentioning. For example, what are the therapeutic molecules and targets etc? Without these information, your sentence "Ongoing pre-clinical research explores the
possibility of targeting lncRNAs using antisense oligonucleotides, CRISPR-based strategies and RNA-based therapies and RNA-based therapies. "in the conclusion and future perspective section reads unsupported.
Comments on the Quality of English Language
acceptable.
Author Response
We thank the reviewer for the suggestion. We have now provided examples of lncRNA-targeting therapeutic strategies and our own perspective in the last paragraph of the revised manuscript. We hope this new addition addresses the reviewers' concern and the manuscript now might be acceptable.